# communications
# engineering

# 300-Gbps optical interconnection using neural-network based silicon microring modulator

Fangchen Hu [1,5], Yuguang Zhang[2,5], Hongguang Zhang[2,5], Zhongya Li[1], Sizhe Xing[1], Jianyang Shi [1], Junwen Zhang [1✉], Xi Xiao[3,4✉], Nan Chi [1✉], Zhixue He[3,4] & Shaohua Yu[3,4]

Silicon microring modulators (Si-MRM) are critical components for high-performance electro-optical (E-O) signal conversion at optical interconnections due to their ultrawide bandwidth. However, the current transmission speed at the interconnections is still limited to 240 Gbps because of the low spectral-efficiency, as a result of the inherent modulation nonlinearity of Si-MRMs. Here, we theoretically analyse the modulation nonlinearity of a depletion-mode Si-MRM. Based on the analytical results, we further propose a physics-inspired neural network, named as bidirectional gate recurrent unit (Bi-GRU) to mitigate the signal distortion in Si-MRMs. Bi-GRU matches the analytical E-O modulation dynamics within Si-MRMs, thus can accurately capture the impairment features and accelerate the data transmission speed. We then fabricate a Si-MRM with −3dB E-O bandwidth of 42.5 GHz, achieving an ultrahigh speed optical interconnection with a data rate of 302 Gbps. The maximum spectral-efficiency of modulated signals is improved to 5.20 bit/s/Hz. The results provide insights to develop ultrahigh-speed Si-MRM using emerging AI techniques.

[1] Key Laboratory for Information Science of Electromagnetic Waves (MoE), Fudan University, Shanghai 200433, China. [2] National Information Optoelectronics Innovation Center, Wuhan 430074, China. [3] State Key Laboratory of Optical Communication Technologies and Networks, China Information Communication Technologies Group Corporation, Wuhan 430074, China. [4] Peng Cheng Laboratory, Shenzhen 518055, China. [5]These authors contributed equally: Fangchen Hu, Yuguang Zhang, Hongguang Zhang. ✉email: junwenzhang@fudan.edu.cn; xxiao@wri.com.cn; nanchi@fudan.edu.cn

Silicon photonics and optical microresonators have enabled important applications in quantum electrodynamics, sensing, optical telecommunications and photonic integrated circuits[1–3]. A critical component in these applications is silicon microring modulator (Si-MRM), which is essential for high-performance electro-optical (E-O) signal conversion. In the field of optical telecommunications, Si-MRM has emerged as a promising E-O modulator candidate for next-generation data center interconnections with ever-increasing requirements for high-speed, low energy consumption, and high-scalability[4–6], due to the advantages of ultrawide bandwidth, low loss, high energy-efficiency (only a few fJ/bit), micron-sized footprint and full complementary metal-oxide-semiconductor compatibility[7].

To realize the high-speed optical interconnections based on Si-MRMs, one challenge is to fabricate a Si-MRM with ultrawide E-O modulation bandwidth. Numerous studies have reported Si-MRMs with E-O modulation bandwidth exceeding 50 GHz[8–10]. However, to the best of our knowledge, the achievable transmission speeds of these ultrawide-bandwidth Si-MRMs have not yet exceeded 240 Gbps until now. The cause of this rate bottleneck is the limited spectral-efficiency (SE) of modulated signals when using Si-MRMs. The inherent free carrier dispersion effect and resonance nature of a Si-MRM distort the waveform of the optical signals after E-O conversion in a nonlinear fashion, thereby leading to the reduction of SE. The highest achievable SE for Si-MRM-based optical interconnections is only 3 bit/s/Hz without applying any digital signal processing methods[11]. For practical communication systems, a high SE is critically important as it determines the limits of data-carrying ability for a given optical bandwidth[12]. Therefore, mitigating modulation nonlinearity and achieving high SE become another challenge for high-speed optical interconnections using Si-MRMs.

Several equalization methods have been proposed to mitigate the modulation nonlinearity at the receiver, such as the traditional Volterra-Wener nonlinear equalizer and support-vector-machine-based methods[13,14]. However, these methods are primarily based on the understanding of static modulation nonlinearity in Si-MRMs induced by their static nonlinear E-O transfer characteristics. The transient modulation nonlinearity of Si-MRMs, which causes overshooting phenomenon[15,16] and signal-level-dependent inter-symbol interference[17] on the modulated signal waveform, has not been adequately addressed. As advanced modulation formats are gradually promoted in high-speed Si-MRM-based optical interconnections[18], the transient modulation nonlinearity becomes more intolerable and urgently requires effective mitigation, given the high sensitivity of high-order modulation formats to nonlinear distortion[19]. Although a pre-distortion circuit has been proposed to partly compensate the signal level-dependent inter-symbol interference, this method only serves for digital modulation formats like pulse-amplitude modulation (PAM)[17], while hardly be promoted to analog modulation formats such as multi-carriers modulation with successive signal levels. Overall, the study of modulation nonlinearity in Si-MRM remains in a preliminary state suffering from a lack of comprehensive theoretical analysis, especially regarding the transient modulation nonlinearity. Here, we theoretically analyze the modulation nonlinearity of a depletion-mode Si-MRM, including the induced distortion on signal waveforms and the dependence of the distortion on signal levels and wavelength detuning ($\Delta\lambda = \lambda_{laser} - \lambda_0$) that defined as the difference between laser wavelength ($\lambda_{laser}$) and the resonance wavelength of a Si-MRM ($\lambda_0$). The existence of the nonlinear distortion on the modulated optical signal is also verified by experiments.

Recent advances in artificial intelligence (AI) have demonstrated its powerful computational modeling ability to obtain intelligent solutions for problems that are virtually impossible to explicitly formulate before[20]. The physics-inspired AI constructed based on known physical mechanisms is preferable over traditional AIs which function like a black box, such as multilayer perception. This preference stems from the fact that users can understand the performance and topology of the physics-inspired AI, relying on the known physical mechanisms, making it more explainable and reliable[21–23]. Building on the concept of physics-inspired AI and our theoretical understanding of the modulation nonlinearity in Si-MRMs, we proposed and demonstrated a powerful, robust, and physics-inspired bidirectional gate recurrent unit (Bi-GRU) neural network with minor modification for the mitigation of the signal distortion in the high-speed E-O modulation of Si-MRMs. The topology of the modified Bi-GRU matches the E-O dynamic modulation model of Si-MRMs, thus effectively capturing features of linear and nonlinear impairments and further accelerating the data transmission speed of Si-MRMs-based optical interconnections.

Based on a self-designed Si-MRM with −3dB E-O bandwidth of 42.5 GHz, we experimentally demonstrate a 302 Gbps O-band optical interconnection and 300 Gbps transmission over a 1-km standard single-mode fiber. The modified Bi-GRU neural network effectively equalizes both the linear and nonlinear impairments on the received signal, providing more than 1.1-dB received optical power (RoP) gain compared to traditional nonlinear equalizers. Moreover, the maximum SE of the transmitted signal is improved up to 5.20 bit/s/Hz thanks to the modified Bi-GRU and the utilization of discrete multitone modulation format with bit and power loading (BPL-DMT). These results enhance the understanding of Si-MRM's high-speed modulation characteristics and confirm the important value of AI technologies in optical communication.

## Results

**Structure and fabrication of Si-MRM**. Using the standard complementary metal-oxide-semiconductor process, we fabricate a high-speed Si-MRM on a 220-nm silicon-on-insulator platform, as shown in the schematic of Fig. 1a. The Si-MRM consists of a bus waveguide and a racetrack ring resonator with the radius ($R$) of 8 μm. To keep the Si-MRM working near the critical coupling region at the wavelength of 1310 nm, the racetrack length and the ring-bus gap width are respectively set to 5 μm and 280 nm to increase the coupling efficiency. Both bus and ring waveguides are ridge waveguides with a width of 420 nm, formed by a 150-nm-depth etch. The slab thickness of the waveguides is designed to be 70 nm for better optical-modes confinement and less bending losses. The ring resonator is composed of a lateral P-N junction whose cross-sectional view is depicted in Fig. 1b. The doping concentration in the P-N junction strongly influences the Si-MRM's E-O bandwidth. Referring to the published results[8], the optimal doping concentration for both P- and N-type doping in our Si-MRM is $3.8 \times 10^{18}$ cm$^{-3}$ to obtain a large E-O bandwidth at a relatively low expense. Since the real resonant wavelength of Si-MRMs always deviates from the predesigned value due to fabrication errors and temperature drift, we integrate a TiN heater with our Si-MRM to effectively control the resonant wavelength based on the thermos-optic effect of silicon. The product of the π-phase-shift voltage and length reaches as high as 0.8 V·cm in our Si-MRM.

**Modulation characteristics of Si-MRMs**. The high-speed E-O modulation of Si-MRM is to exploit the plasma dispersion effect, in which the concentration of free charges in silicon changes the real and imaginary parts of the refractive index[24]. In this way, the phase and intensity of output optical field ($E_{out}$) can be tunable (modulated) by electrical manipulation of the charge density in the P-N junction of Si-MRMs, such as carrier injection or depletion by applying a forward or reverse electrical bias.

As shown in Fig. 2, we investigate the signal modulation characteristics of our Si-MRM. The E-O transfer curve, which plots

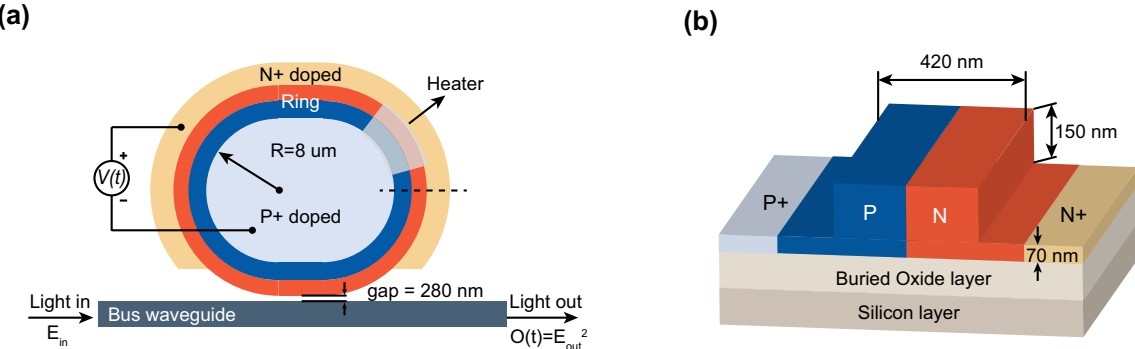

**Fig. 1 Schematic and structure of the self-designed racetrack Si-MRM. a** The incident light into the bus waveguide has the optical field Ein. The electrical signal V(t) to be transmitted is applied to the P-N junction of Si-MRM and finally modulated to the output optical signal O(t) whose intensity is equal to the square of output optical field Eout. *R*: radius. **b** A 3-D cross-sectional view of the P-N junction at the dash line in **a**.

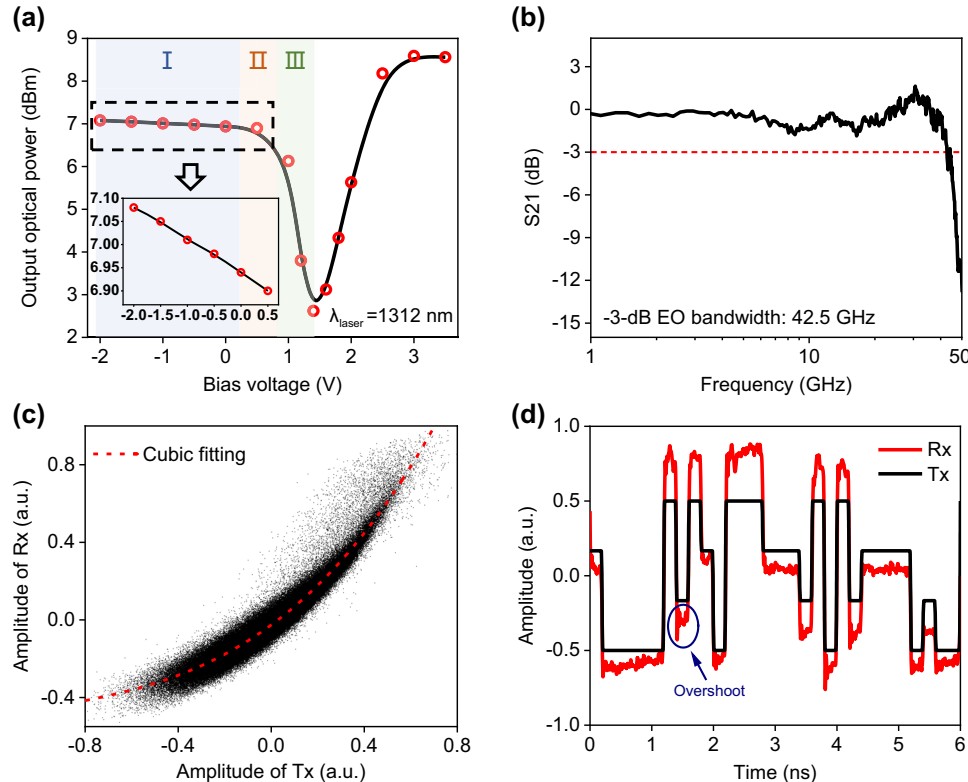

**Fig. 2 Modulation characteristics of the Si-MRM. a** E-O modulation transfer curve (output optical power versus bias voltage). Inset: the amplified picture of the E-O modulation transfer curve when bias voltage ranging from −2 V to −0.5 V. I: linear modulation region. II: Nonlinear modulation region. III: High-extinction-ratio modulation region. **b** The −3dB E-O bandwidth at the bias voltage of −0.9 V. **c** Static modulation nonlinearity indicated by the measured Amplitude/amplitude curve of transmitted (Tx) and received (Rx) electrical (bias voltage is −0.9 V and Vpp of signal is 0.6 V). **d** The normalized waveforms of 5-GBaud PAM-4 transmitted (Tx) and received (Rx) signals (bias voltage is −0.9 V and Vpp of signal is 0.2 V). The mazarine circle marks the overshoot phenomenon, i.e., the transient modulation nonlinearity. Laser wavelength ($\lambda_{laser}$) in all figures is 1312.0 nm.

the dependence of the output optical power on the bias-voltage, shows a nonlinear Lorentz-line shape when the input laser wavelength is set to 1312 nm (Fig. 2a). The Si-MRM reaches its critical resonance point, where it outputs the lowest optical power (~2.5 dBm), at ~1.5 V bias voltage. Additionally, the output optical powers at negative and positive bias voltages are asymmetric due to the differences in working modes: the carrier depletion and injection mode, respectively, which result in different attenuation for the transmission of light. We manually divide the modulation region where the bias voltage can be set into three regions: linear region (I), nonlinear region (II) and high-extinction-ratio region (the insets in

Fig. 2a). There exists a trade-off among modulation characteristics (RoP, extinction-ratio, modulation bandwidth and nonlinearity) at different modulation regions. Given that the carrier-depletion mode can achieve larger E-O modulation bandwidth than the carrier-injection mode[25], applying a negative bias voltage (Region I) for signal loading is better than a positive one for high-speed signal modulation. Furthermore, Region I also has the highest RoP, though its extinction-ratio is lower than other regions. For short-reach optical interconnection scenarios where no optical amplifiers are applied, the optical signal with a higher RoP is crucial for higher data rate similar as extinction-ratio. Hence, the bias voltage needs to be

optimized by parameters iteration in experiments to balance the trade-off among these signal modulation characteristics. The combined effects of these characteristics eventually determine how fast the optical interconnection can be achieved. Since the essence to affect the modulation characteristics of bias voltage is that it changes the wavelength detuning, the laser wavelength which changes the wavelength detuning as well also needs iterative optimization. The experimental results of the parameters iteration (Supplementary Fig. S1) show that the optimal bias voltage is −0.9 V at the optimal laser wavelength of 1312 nm. The input laser power is 15 dBm, the output optical power of the Si-MRM is >6.9 dBm at the optimal parameters. The measured S21 parameter is shown in Fig. 2b, from which the -3-dB E-O bandwidth is around 42.5 GHz (measurement method is seen in Methods).

The modulation nonlinearity is studied in Fig. 2c, d, including the static and transient nonlinearity. Measuring the amplitude/amplitude curve of transmitted and received signals is a typical method to show the static nonlinearity of a communication system (The measurement method is given in **Methods**). Figure 2c presents the amplitude/amplitude curve when transmitting a 5-Gbaud 4-levels pulse-amplitude-modulation (PAM-4) signal at a back-to-back case. A cubic nonlinearity is observed after polynomial fitting, which reflects the static nonlinearity of our Si-MRM. Moreover, there exists overshoot phenomenon (marked by a mazarine circle in Fig. 2d) in the waveforms of the PAM-4 received signal, which can be classified into a type of transient modulation nonlinearity since it only appears at the signal edge where the signal level starts to change. The induced static and transient nonlinear signal distortions will cause errors during the process of signal demodulation and symbol decision. To better mitigate these nonlinear signal distortions, a theoretical analysis of modulation nonlinearity in Si-MRMs need to be established to guide the design of effective nonlinear equalizers.

**Theoretical analysis of modulation nonlinearity in Si-MRMs.** Let $a(t)$ be the partial energy of the incident light entering the ring resonator from the bus waveguide $E_{in}(t) = E_0 e^{j\omega t}$ ($E_0$: amplitude, $\omega$: angular frequency) with the mutual parameter $\mu$. The dynamic characteristics of the energy flowing in the Si-MRM

can be described by the coupled mode equation[26]:

$$\frac{d}{dt}a(t) = \left(j\omega_r - \frac{1}{\tau}\right)a(t) - j\mu E_{in}(t) \tag{1}$$

where the resonance frequency of Si-MRM $\omega_r$ and the power decay rate constant $\frac{1}{\tau} = \frac{1}{\tau_e} + \frac{1}{\tau_l}$ are controlled by the input electrical signal $V(t)$. The $\tau_e$ and $\tau_l$ are time constants of the power decay due to ring resonance losses and coupling losses, respectively. The output light $E_{out}(t)$ can be connected with $E_{in}(t)$ by Eq. (2):

$$E_{out}(t) = E_{in}(t) - j\mu a(t) \tag{2}$$

The E-O signal conversion in Si-MRM can be numerically simulated by Eqs. (1) and (2) using a split-step iterative method over a small time-step $\Delta t = t_{k+1} - t_k$ [27]. Assuming the electrical signal $V_k$ at the $k$th small time slot is constant, the iterative equation of output optical signal $O_k(t) = E_{out}^2(t)$ at the $k$th time slot ($t_k \leq t \leq t_{k+1}$) is given:

$$\frac{E_{in} - \sqrt{O_k(t)}}{j\mu}$$
$$= \underbrace{\left[\frac{E_{in} - \sqrt{O_{k-1}(t_k)}}{j\mu} - Q_k(V_k)\exp(j\omega t_k)\right]\exp\left[\left(j\omega_{r,k}(V_k) - \frac{1}{\tau_k(V_k)}\right)(t - t_k)\right]}_{D_k(t)}$$
$$+ \underbrace{Q_k(V_k)\exp(j\omega t)}_{S_k(t)} = D_k(t) + S_k(t) \tag{3}$$

Where $Q_k = -j\sqrt{\frac{2}{\tau_e(V_k)}}/[j(\omega - \omega_r(V_k)) + \frac{1}{\tau(V_k)}]$. Based on Eq. (3), the simulation of the E-O modulation process of a multiple-level input electrical signal $V(t)$ is depicted and analyzed in Fig. 3a–d. The $V(t)$ goes through the static nonlinear response ($S(t)$) and the transient nonlinear response ($D(t)$) before converted to the $O(t)$. The simulation results of the nonlinear distortion of $S(t)$ and $D(t)$ are consistent with the experimental results analyzed above, which leads to nonlinear level distribution and overshoot, respectively.

The overshoot results in the optical responses ($O_k(t_{k+1})$ and $O_{k-1}(t_k)$) of $V_k(t_{k+1})$ and $V_{k-1}(t_k)$ which are equal originally,

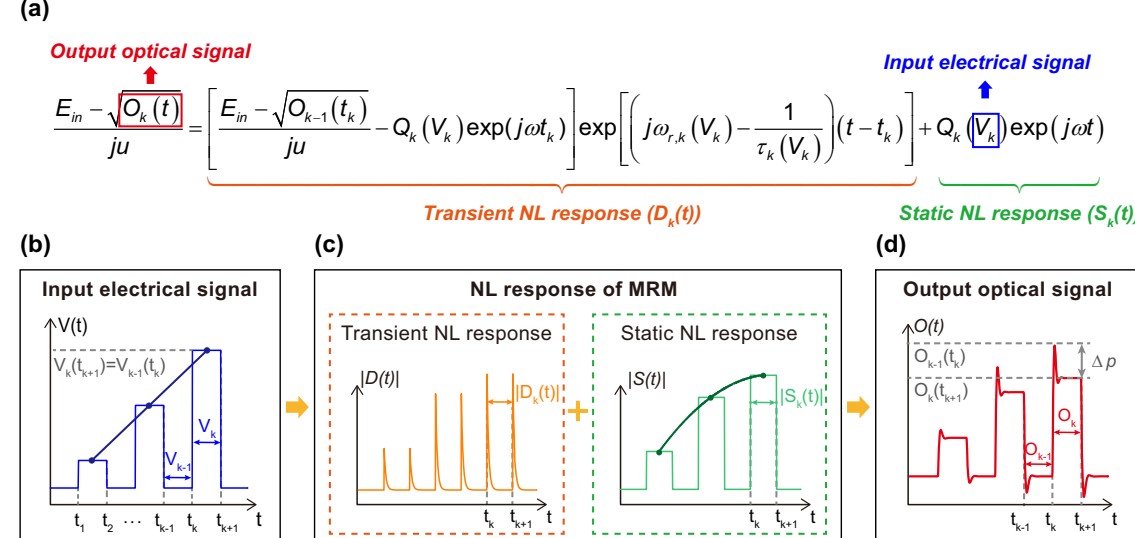

**(a)**

*Output optical signal*          *Input electrical signal*

$$\frac{E_{in} - \sqrt{O_k(t)}}{ju} = \underbrace{\left[\frac{E_{in} - \sqrt{O_{k-1}(t_k)}}{ju} - Q_k(V_k)\exp(j\omega t_k)\right]\exp\left[\left(j\omega_{r,k}(V_k) - \frac{1}{\tau_k(V_k)}\right)(t - t_k)\right]}_{\text{Transient NL response } (D_k(t))} + \underbrace{Q_k(V_k)\exp(j\omega t)}_{\text{Static NL response } (S_k(t))}$$

**(b)** Input electrical signal  **(c)** NL response of MRM  **(d)** Output optical signal

**Fig. 3 Simulation results and analysis of the modulation nonlinearity in a depletion-mode Si-MRM. a** Theoretical model for E-O intensity modulation, described by an iteration equation correlating the output optical signal ($O_k(t)$) and the input electrical signal ($V_k$) in the $k$th time slot ($t_k \leq t \leq t_{k+1}$). **b–d** The linearly distributed electrical signal $V(t)$ experiences transient nonlinear (NL) response ($D_k(t)$) and static NL response ($S_k(t)$) before turning to optical signal $O(t)$. The two NL responses lead to explicit NL distortions on the waveform of $O(t)$: an overshoot phenomenon and nonlinear level distribution. The $V(t)$ with larger wavelength detuning (indicating a higher amplitude here) has a higher overshoot amplitude ($\Delta p$) in $O(t)$.

become unequal with the amplitude difference of $\Delta p$. The distortion degree of the transient nonlinearity on signals can be quantified by the magnitude of $\Delta p$. Obviously, a bigger $\Delta p$ means a higher possibility to cause signal decision errors. By simulation (Supplementary Fig. S2 and Note 1), it's found that the magnitude of $\Delta p$ at the $k$th time slot depends on the levels and wavelength detuning of input electrical signals at $k$th and $(k-1)$th time slots. Specifically, a bigger $\Delta p$ appears at the $(V_k, V_{k-1})$ with larger amplitudes and wavelength detuning. For example, the $\Delta p$ of $(-5\,\mathrm{V}, -7\,\mathrm{V})$ is bigger than that of $(-3\,\mathrm{V}, -5\,\mathrm{V})$. This dependence can also be extracted from Eq. (3) in which that $O_k(t)$ is determined by $V_k$ and the optical response at the last time slot $(O_{k-1}(t_k))$ which in turn is determined by $V_{k-1}$ and $O_{k-2}(t_{k-1})$. Finally, we can get that $O_k(t)$ depends on $[V_k, V_{k-1}, ..., V_1]$ if we continue with this recurrence. Both static and transient nonlinear distortions are experimentally demonstrated and highlighted in the eye diagram of a 5-Gbaud PAM-4 received signal (Supplementary Fig. S3).

**Physics-inspired AI-accelerated ultrahigh-speed Si-MRM-based optical transmission.** Based on the above insights into the modulation nonlinearity of Si-MRM, we design a modified Bi-GRU neural network and then achieve an ultra-high-speed Si-MRM-based optical interconnection accelerated by the artificial neural network. The experimental setup is shown in Fig. 4a with detailed introduction in Methods. To maximize the SE and approach the Shannon limit, the BPL-DMT modulation is applied to adapt the uneven channel and efficiently allocate bits in frequency subcarriers[28,29]. At the receiver, a modified Bi-GRU neural network is used to reconstruct the transmitted signal $V(t)$ from the received signal $O(t)$ suffering from linear and nonlinear impairments before the normal BPL-DMT demodulating process (Fig. 4b). The specific digitial signal processing (DSP) for signal modulation and recovery are seen in Methods. The modified Bi-GRU neural network consists of three layers: an input layer, a hidden layer and an output layer. The hidden layer is composed of two GRU cells with different time-flow directions. Different from traditional Bi-GRU neural network that two directional GRU cells use identical input data samples, the modification is that the data entering the forward GRU is the received signal from $(k-d)$th to $k$th time slots, and the data entering the backward GRU is the received signal from $k$th to $(k+d)$th time slots. Their final memory state vectors $C_k^f$ and $C_k^b$, respectively coming from the forward and backward GRUs, cascade together into one vector flowing into the output layer.

Rather than a "black box" application of AI models driven by hundreds of samples, the inner configuration of our modified Bi-GRU neural network is explainable and matches the modulation model of Si-MRMs. As shown in Fig. 4c, the modulation model of a Si-MRM is that the $k$th output optical signal $O_k$ has a time-relevant pattern dependency on previous optical responses $O_{k-1}$ and the $k$th input electrical signal $V_k$, given as $O_k = \Im(O_{k-1}, V_k)$ in Eq. (3), where $\Im(\cdot)$ is the dynamic nonlinear response of a Si-MRM. Referred to the equalization mechanism of traditional Volterra filters, an equalizer having a similar model with the physic system can effectively approximate the inversion of the

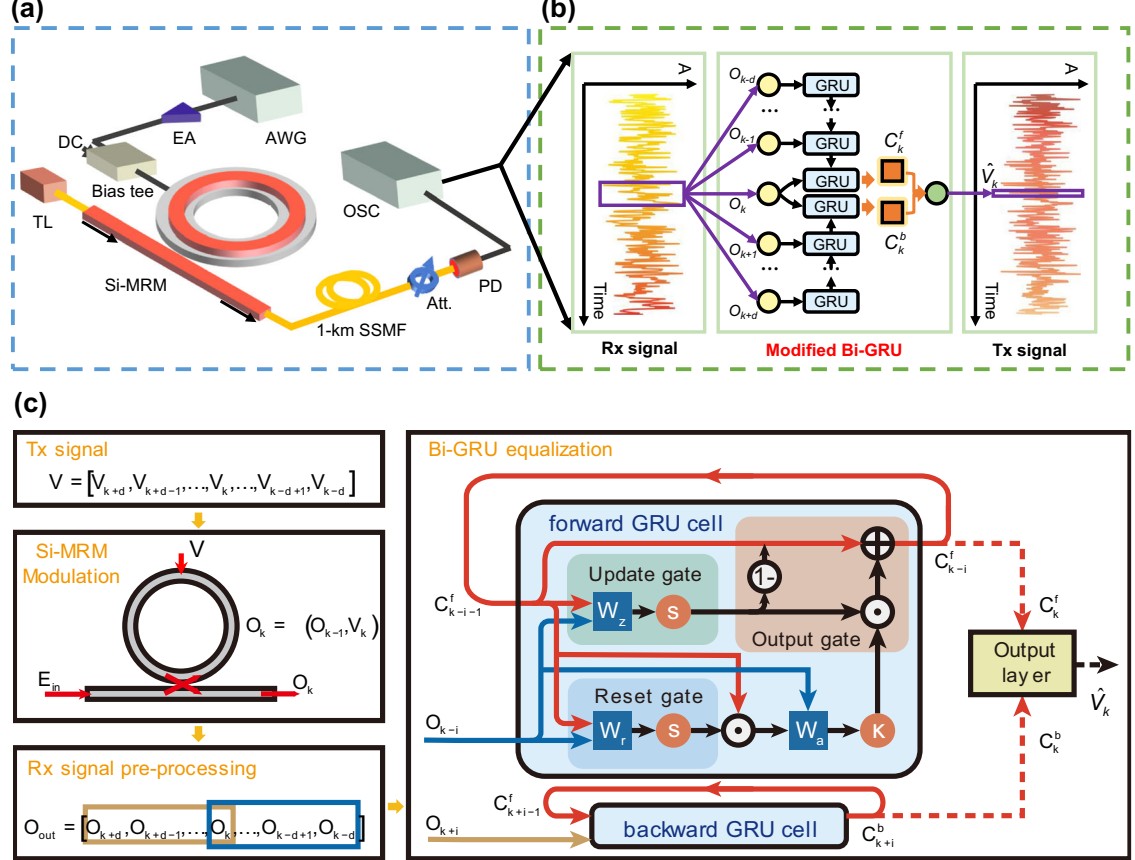

**Fig. 4 The proposed AI-accelerated Si-MRM-based optical transmission framework. a** The proof-of-concept experimental system to realize high-speed short-reach optical interconnection. (AWG arbitrary waveform generator, EA electrical amplifier, DC direct current, TL tunable semiconductor laser, SSMF standard single-mode fiber, Att. attenuator, PD photodetector, OSC oscilloscope). **b** The modified bidirectional gate recurrent unit (Bi-GRU) neural network is used to reconstruct the transmitted signal (Tx signal) from the received signal (Rx signal) captured from the OSC. (A: Amplitude). **c** The principle of Bi-GRU serving as a post-equalizer and the inner configuration of a GRU cell.

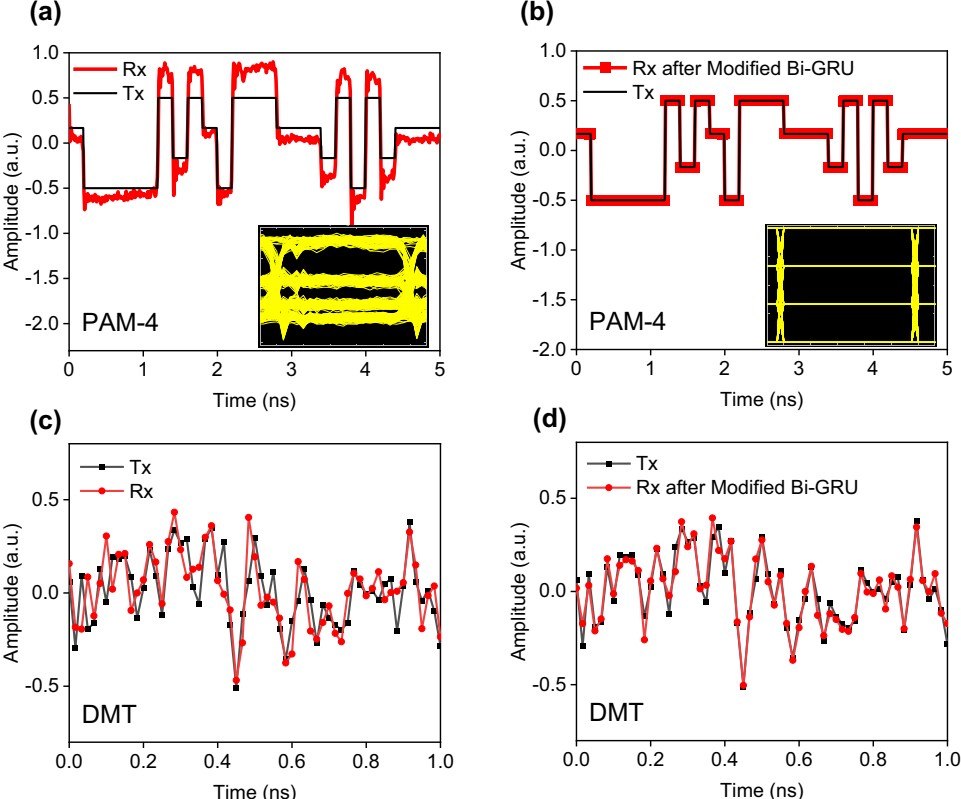

**Fig. 5 Equalization performance of the modified Bi-GRU neural network for PAM-4 and DMT received signal. a** The waveforms of transmitted (Tx) and received (Rx) PAM-4 signal at the sample rate of 5-GBaud. After the equalization of the modified Bi-GRU neural network on the received signal, the corresponding waveforms (red line) are shown in **b**. Insets are their eye-diagrams. For 120-GBaud DMT signal, the waveforms of transmitted, received and equalized signal are shown in **c** and **d**.

system by the fixed-point approach[30]. Here, our proposed forward GRU just provides such a similar time-relevant model as the modulation model of a Si-MRM. In the inner structure of the forward GRU, the $(k-i)_{th}$ memory state $C_{k-i}^f$ depends on previous memory states $C_{k-i-1}^f$ and the $(k-i)_{th}$ input sample $O_{k-i}$, expressed as $C_{k-i}^f = \aleph(C_{k-i-1}^f, O_{k-i}) \, (0 \le i \le d)$, where $\aleph(\cdot)$ is the inner nonlinear structure of a GRU cell (Supplementary Information Note. 2). Therefore, the forward GRU theoretically has an effective nonlinear equalization ability to the modulation nonlinearity of Si-MRMs. Except for the nonlinear distortions, the mitigation of the inter-symbol interference effect resulting from the bandwidth limitation of Si-MRM and other E-O devices, requires the equalizer to consider the influence of symbols in future time slots. Hence, a backward GRU cell involved with the received symbols from $k_{th}$ to $(k + d)_{th}$ time slots is merged with the forward GRU into the final modified-Bi-GRU. The effective nonlinear and linear equalization performance of the modified Bi-GRU is demonstrated experimentally shown in Fig. 5a, b for PAM-4 signals and Fig. 5c, d for BPL-DMT signals.

Besides the advantage of interpretability, the proposed modified Bi-GRU also improves the equalization performance and convergence speed compared to a fully-connected neural network which is a typical representative of the data-driven AI technology. In Fig. 6a, the fully-connected neural network requires more than 140 epochs to decrease the bit error rate (BER) below the 20% soft-decision forward-error correction (SD-FEC) threshold. On the contrary, our modified Bi-GRU only needs 40 epochs reflecting a faster convergence speed. Furthermore, we verify that the proposed modified Bi-GRU effectively captures channel response instead of the generation pattern of the pseudo random bit sequence (PRBS).

Three patterns of PRBS are generated by Mersenne Twister algorithm[31] that has the period of $2^{19937}$ -1. The 30% of Pattern 1 is used to train the modified Bi-GRU, the others (Pattern 2 and 3) are used for BER test. The results in Fig. 6b means the modified Bi-GRU accurately learns the information of channel and has a robust equalization ability for different data patterns.

A demonstration of a ultrahigh-speed transmission experiment based on the modified Bi-GRU neural network is carried out. We first evaluate the BER performance of the modified Bi-GRU neural network compared to that of other traditional linear and nonlinear equalizers at different RoPs. The preset data rate of the transmitted BPL-DMT-modulated signal is 280 Gbps with a baud rate of 60 Gbaud. The linear equalizer utilized for comparison is a feedback forward equalizer based on the recursive least squares algorithm. The nonlinear equalizer is the third-order polynomial nonlinear equalizer (PNLE) based on the recursive least squares algorithm, which has sufficient equalization ability for third-order static nonlinear impairment. The results in Fig. 7a show that both the PNLE and modified Bi-GRU greatly outperform the linear RLS equalizer, which means that nonlinear distortion occupies a significant part of the whole signal distortion. Compared to the PNLE, the modified Bi-GRU provides an ~1.1-dB gain of the RoP under the 20% SD-FEC threshold because the modified Bi-GRU considers the extra transient nonlinear impairments and has better optimization ability. The BER performance for transmitting over 1 km shows a similar level to that of the back-to-back case, which verifies the ability of the Si-MRM for high-speed short-reach optical interconnection. To obtain the maximum data rate achievable by the Si-MRM, we test the data rate under different baud rates when the RoP is set to 7 dBm (Fig. 7b). The data rate at the back-to-back could reach 302 Gbps at a baud rate

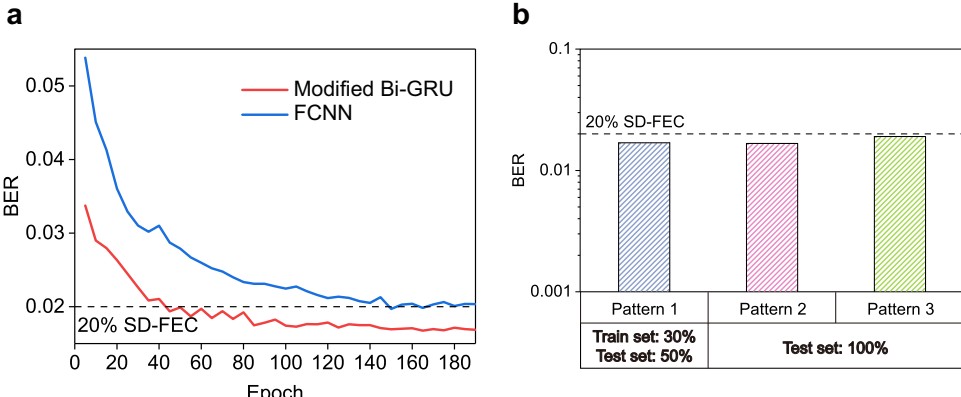

**Fig. 6 Convergence speed and robustness analysis of the modified Bi-GRU. a** Convergence curve of the BER when using the modified Bi-GRU and fully-connected neural network (FCNN) to equalize received signal at a data rate of 302 Gbps. The modified Bi-GRU has a better equalization performance and faster convergence speed. **b** BER of three groups of received signal sequences with different generation patterns, after the equalization of a Bi-GRU with fixed neural network structure.

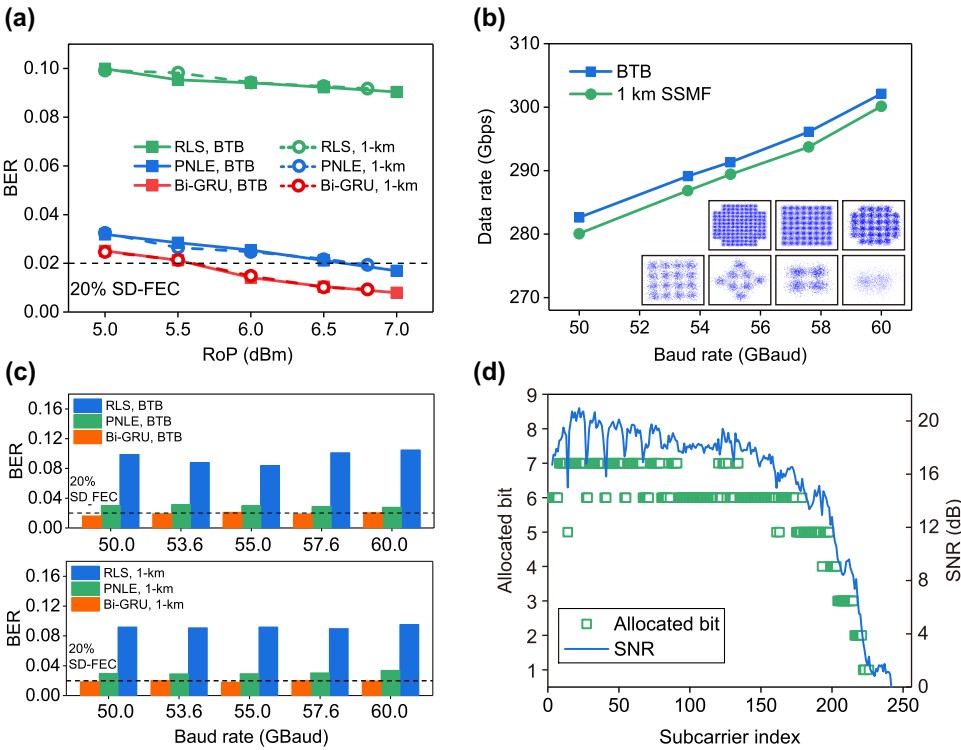

**Fig. 7 Experimental results of the proof-of-concept Si-MRM-based optical interconnection. a** BER performance of three equalization schemes versus different received optical power (RoP) at a data rate of 280 Gbps. (RLS: a feedback forward equalizer based on recursive least squares algorithm. PNLE: 3rd-order polynomial nonlinear equalizer). **b** Data rate versus baud rate in the cases of back-to-back (BTB) and 1 km SSMF. Insets: constellation diagrams of the received QAM signal at a data rate of 302 Gbps. **c** BER performance of three equalization schemes versus different baud rates for the BTB and 1 km fiber transmission cases. **d** The SNR distribution and bit allocation at different subcarriers when the data rate reaches 302 Gbps.

of 60 Gbaud with an SE of ~5.20 bit/s/Hz after removing the redundancy of 8 zero-padded subcarriers ($data\ rate = \frac{SE \times B \times (Fs - 2z)}{Fs}$). The $B$, $Fs$ and $z$ present baud rate, the size of fast Fourier transform and the number of zero-padded subcarriers, respectively. Here, $Fs = 512$ and the first two subcarriers and last six subcarriers are abandoned ($z = 8$). The maximum data rate over the 1-km standard single-mode fiber is 300 Gbps with an SE of ~5.16 bit/s/Hz. The BER performance of the three equalization schemes shown in Fig. 7c further verifies the powerful resistance of the modified Bi-GRU to the linear and nonlinear impairments existing in the Si-MRM. Only the

modified Bi-GRU could decrease the BER to below 2e−2 to help achieve 302 Gbps. Figure 7d shows the SNR and bit distribution of the DMT signal at a data rate of 302 Gbps. With the aid of the modified Bi-GRU and the bit and power loading technology, bits were adaptively allocated to every subcarrier. The low-frequency subcarriers could support 32- to 128-quadrature amplitude modulation (QAM), while high-frequency subcarriers (around 50 GHz, i.e., the 220th subcarrier) only allow binary-phase-shift-keying modulation on signals. One sets of constellation diagrams of the signal at the rate of 302 Gbps are presented as insets in Fig. 7b.

## Conclusions

To improve the Si-MRM performances, we comprehensively optimize the Si-MRM from the device, to modulation as well as the signal processing. We fabricate a racetrack Si-MRM with a -3-dB electrical bandwidth of 42.5 GHz enabled by doping concentration optimization. Although the bandwidth is greatly extended, nonlinear impairments induced by the Si-MRM dramatically restrict the overall modulation capacity. To fully compensate for this, we theoretically analyze the mechanism of two kinds of nonlinear impairments and their time-relevant dependency on signal levels and wavelength detuning. This analysis helps to fully understand the modulation nonlinearity in Si-MRM, and establishes the connection between the physical model and advanced signal processing technologies such as AI. Consequently, a Bi-GRU neural network with minor modification based on the obtained physical knowledge of Si-MRM's modulation nonlinearity is applied to help accelerate the Si-MRM modulation speed. We experimentally demonstrate that by selecting a suitable configuration of neural networks according to the dynamic nonlinear response of the Si-MRM, the AI technology can accelerate the modulation of the Si-MRM to a record-breaking data rate of 302 Gbps. The Bi-GRU neural network presents a better equalization performance than traditional nonlinear equalizer. The results in this work offer deeper insight into DSP for Si-MRM-based high-speed optical interconnections, validating the concept that the AI technologies based on the deep understanding of physics system could further improve the modulation capacity of Si-MRMs. The concept could also be promoted to other disciplines and facilitate their development.

## Methods

**Experimental setup**. The arbitrary waveform generator (AWG, Keysight M8194A) first generates the signal stream to be transmitted, which is then amplified by an electrical amplifier (SHF S807C, 55 GHz). Next, a wide-bandwidth bias tee (SHF BT65B) imposes the amplified signal on a bias voltage to drive the P-N junction of the Si-MRM by a high-frequency ground-signal radio-frequency probe. The bias voltage is supplied by a direct current (DC) power supply (KEITHLEY 2230G-30-1). The optical carrier is provided by a tunable semiconductor laser (TSL, Santec TSL-550) with a laser wavelength of 1312.0 nm and coupled into the bus waveguide through a fiber patch cord and a grating coupler. The coupling efficiency for the transverse electric (TE) mode increases to 40% with the grating coupler, which is also used in the distal end of the bus waveguide. The output optical signal is converted to an electrical signal by a photodetector (PD, FINISAR XPDV2320R) after transmitting over a 1-km SSMF and an optical attenuator (Att.). The electrical signal is captured by an oscilloscope (KEYSIGHT UXR0704A), which has a maximum sample rate of 256 GS/s.

**E-O bandwidth and modulation nonlinearity measurement**. The −3dB E-O bandwidth of the Si-MRM can be obtained by measuring the S21 parameters of the whole communication system using the inner application software in AWG and oscilloscope. Since the bandwidth of other electron and photoelectric devices except the Si-MRM is wider than that of the Si-MRM, the obtained bandwidth of the whole communication system can be equivalent to the E-O bandwidth of the Si-MRM.

A typical characterization of the static E-O modulation nonlinearity of Si-MRMs is measuring amplitude/amplitude curve, in which the x-axis is the amplitude of the transmitted electrical signal sequence sending to AWG, the y-axis is the amplitude of the correspondingly received electrical signal that captured by OSC and then directly resampled to the same baud rate with the transmitted signal. The transmitted signal is a 5-Gbaud PAM-4 modulated digital signal. Both the transmitted and received electrical signals are normalized to $[-1, 1]$. If there is no modulation nonlinearity during the E-O conversion of the Si-MRM, the amplitude/amplitude curve will present a linear trend. Otherwise, the curve will be nonlinear. It's noted that the RoP is adjusted to avoid the nonlinear region of PD.

**DSP for the ultrahigh-speed optical transmission experiment**. The flow diagram of the DSP procedure for the 302 Gbit/s optical transmission experiment is presented in Supplementary Fig. S4, which consists of two steps. The first step is to estimate the SNR at every subcarrier of the DMT signal. Then, the estimated SNR guides the second step, i.e., bit and power loading, to adaptively allocate the optimal bit and power on every subcarrier. In the first step, we generate a random PRBS (period: $2^{19937}$ -1) based on the Mersenne-twister algorithm[31], which is then mapped to a QAM-4 signal serial sequence. After being converted to a parallel sequence, the 4-QAM signal is modulated to a DMT signal with a fast Fourier transform size of 512 according to the standard DMT modulation process[32]. The cyclic prefix (CP), with a length of 16, is added to the head of every DMT symbol to eliminate the inter-symbol interference. The parallel DMT signal is then transferred to a serial DMT signal, which is the ultimate training signal fed into the AWG. The maximum amplitude of the DMT signal is normalized to a unit value. Considering the poor response at low and high frequencies of the channel, the first two subcarriers and last six subcarriers are unused here. At the receiver, the captured signal is first synchronized and resampled to the same baud rate as that of the transmitted signal. Then, the Bi-GRU is used to equalize the linear and nonlinear impairments of the signal. After that, the equalized signal is transferred to the frequency domain by the inverse fast Fourier transform operation. The residual linear distortion of each subcarrier on the signal is equalized by the zero-forcing equalizer (ZFE) and intra-symbol frequency-domain averaging (ISFA) technologies. The sampling frequency offset (SFO) is mitigated using two DMT symbol pilots that lie on the head and tail of the DMT signal sequence introduced in ref. [33]. Subsequently, the error-vector magnitude of the signal is used to estimate the SNR of every subcarrier under the preset 20% SD-FEC BER threshold of 2E-2[34,35]. It produces an optimal QAM order and power ratio allocation strategy for each subcarrier based on the Levin-Campello algorithm[28,36]. In the second step, another group of PRBSs is modulated to a new DMT signal sequence based on the allocation strategy obtained in the first step. The modulation and demodulation process and DSP process are the same as those in the first step. The BER is calculated by using the recovered PRBS and the transmitted PRBS.

**Parameters of the Bi-GRU neural network**. The ratios of the training set, validation set and test set are 30%, 20% and 50%, respectively. To achieve the best equalization performance and decrease the parameter complexity as much as possible, the optimal length of time windows and optimal dimensions of the memory state are 11 and 80, respectively, obtained by parameter iteration optimization. The batch size and training epoch are set to 64 and 60, respectively, to reduce the amount of time spent on back-propagation.

## Data availability

The data that support the findings of this study are available from the corresponding author upon reasonable requests.

## Code availability

The relevant code is publicly available at: https://github.com/huffff/Modified_Bi_GRU.

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

## Acknowledgements

This work was partly support by National Key Research and Development Program of China (2022YFB2802803 (N.C.)), NSFC project (No.61925104 (N.C.), No.62171137 (J.W.Z.)), Shanghai NSF project (No. 21ZR1408700 (J.W.Z.)) and The Major Key Project of PCL (N.C.).

## Author contributions

F.C.H., Y.G.Z. and H.G.Z. contributed equally to this work. F.C.H., H.G.Z., Y.G.Z., Z.Y.L., S.Z.X and J.W.Z. conducted the experiments. F.C.H., Z.Y.L., J.Y.S., N.C. and J.W.Z. contributed to the theoretical analysis and simulation. F.C.H. analyzed the data and wrote the paper. Y.G.Z., H.G.Z. and X.X. fabricate the Si-MRM and establish the experimental setup. Z.X.H., N.C. and S.H.Y. supervised the research project. All authors commented on the manuscript.Data availability
The data that support the findings of this study are available from the corresponding author upon reasonable requests.

## Competing interests

The authors declare no competing interests.
