## [Peer Review File · Communications Engineering]

Reviewers' comments:

Reviewer #1 (Remarks to the Author):

In this paper, the authors showed equalization method with physics-inspired neural network to improve the spectral efficiency of Si micro-ring modulator. Proposed signal processing method mitigates modulation non-linearity-driven waveform distortions at the receiver side. By using this technique authors have achieved 302 Gbps transmission speed.

The authors presented some interesting approach and results. However, they might want to rewrite the paper or present the results in a different way.

Major comment:

Comment #1:

There is inconsistency between wavelength used in demonstration and spectrum measurement in supplementary information. 1312nm is used in whole demonstration, but in the Fig.S1 there is no spectrum result on 1312nm. In Fig.S1, authors showed resonance near 1310nm and based on their device description, next resonance should be around 1317nm. These resonance peaks are not closer to 1312nm at all, and this is big misleading. I understand this might coming from the fact that Fig.S1 is simulation. Personally, I would recommend adding measured spectrum directly in manuscript.

Comment #2:

Line 158, authors mention optimal bias voltage of -0.9 V, with one level $V_{pp}=0.2$ V and 3-level $V_{pp}=0.6$ V (mentioned in Fig. 2 caption). However, in Fig. 2a we see that curve is quite flat around -0.9 V. It makes more sense to modulate around +0.9 V, since this value is on the slope of curve, which will produce much higher extinction ratio. Is x-axis (bias) any how reverted in graph Fig. 2a? Authors also stress they drive modulator in reverse bias (depletion mode). Sketch below explains the concern:

Comment #3:

I would like to recommend adding pam4 eye-diagram in the manuscript if they have in addition to Fig.5. Would be more impactful to show how eye-diagram can be improved through modified Bi-GRU

Comment #4:

Please add PRBS length information. PRBS7 vs PRBS31 can make quite different result in BER measurement. Also I would like to recommend to add optical power and loss over the link.

Minor comments:

Comment #1:

Line 127: Figure 2a caption mentions inset (the amplified picture of the E-O modulation transfer curve when bias voltage ranging from -2 V to -0.5 V) which is absent in the figure itself. Please insert it.

Comment #2:

Line 130: word signal is missing after word electrical.

Comment #3:

Lines 127-132: 1312 nm wavelength is mentioned several times. Since all figures show results with this wavelength, please mention it just once in the caption.

Comment #4:

Line 224: In Figure 4a there is a TL acronym, but in figure caption it is mentioned as TSL (tunable semiconductor laser). Please keep consistency.

Reviewer #2 (Remarks to the Author):

The authors have made much efforts to increase the modulation speed of silicon MRR via device design, modulation as well as signal processing. The key novelties they claim include two points, i.e. theoretical analysis of modulation nonlinearity in Si-MRMs, as well as design a physics-inspired bidirectional gate recurrent unit (Bi-GRU) neural network. Finally, they demonstrated optical interconnections at data rate up to 302 Gbps, showing a 1.1-dB received optical power gain than traditional nonlinear equalizers with spectral-efficiency of modulated signal to 5.20 bit/s/Hz. Although the authors managed to provide solid experimental results, the connection between two novelties is not well revealed. The authors have tried to describe the motivation of constructing a physics-inspired AI network, which to be best of own understanding, is to take the real modulation process into consideration. This is very interesting and important, but unfortunately not well explained, at least a better way should be find to explain the connection between physics-inspired AI (Fig. 4c) and theoretical model, to aid understanding from people who are not familiar with processes of neuron networks. From another point of view, the novelty of this work is not sharp. The theoretical analysis of modulation nonlinearity serves as a tool to help developing advanced AI network, and therefore how the physics-inspired AI network is constructed and why it provide great performance should be elaborated. Apart from this general point, I has a few other suggestions that the authors may consider to improve the work. Among these issues, the most critical part is that, the current manuscript has too many serious grammar errors, which obviously can not be acceptable for the publication in Communications Engineering. In conclusion, I suggest major revision of this work, both technically and grammatically.

1.Line 112-114: the authors claim that “the radius (R) and racetrack length of the ring resonator is 8 um and 5 um, respectively, to keep the Si-MRM working near the critical coupling region at the wavelength of 1310 nm.” In my opinion, the critical coupling condition of an MRR has no direct connection with the radius and racetrack length.

2.How was Fig. 2c measured? This should be described.

3.Line 210-211: what is “wavelength detuning”? What is “wavelength difference”?

4.Just as an concrete example, a serious grammar error occurs at line 206-207: Since equation (3) tells .. is decided by the V_k and optical response at last time step ... which in turn relates to ... and further previous optical response. The sentence is not complete.

Reviewers' Comments:

Reviewer #1 (Remarks to the Author):

In this paper, the authors showed equalization method with physics-inspired neural network to improve the spectral efficiency of Si micro-ring modulator. Proposed signal processing method mitigates modulation non-linearity-driven waveform distortions at the receiver side. By using this technique authors have achieved 302 Gbps transmission speed.

The authors presented some interesting approach and results. However, they might want to rewrite the paper or present the results in a different way.

Major comment:

Comment #1:

There is inconsistency between wavelength used in demonstration and spectrum measurement in supplementary information. 1312nm is used in whole demonstration, but in the Fig.S1 there is no spectrum result on 1312nm. In Fig.S1, authors showed resonance near 1310nm and based on their device description, next resonance should be around 1317nm. These resonance peaks are not closer to 1312nm at all, and this is big misleading. I understand this might coming from the fact that Fig.S1 is simulation. Personally, I would recommend adding measured spectrum directly in manuscript.

Comment #2:

Line 158, authors mention optimal bias voltage of -0.9 V, with one level $V_{pp}=0.2$ V and 3-level $V_{pp}=0.6$ V (mentioned in Fig. 2 caption). However, in Fig. 2a we see that curve is quite flat around -0.9 V. It makes more sense to modulate around +0.9 V, since this value is on the slope of curve, which will produce much higher extinction ratio. Is x-axis (bias) any how reverted in graph Fig. 2a? Authors also stress they drive modulator in reverse bias (depletion mode). Sketch below explains the concern:

Comment #3:

I would like to recommend adding pam4 eye-diagram in the manuscript if they have in addition to Fig.5. Would be more impactful to show how eye-diagram can be improved through modified Bi-GRU

Comment #4:

Please add PRBS length information. PRBS7 vs PRBS31 can make quite different result in BER measurement. Also I would like to recommend to add optical power and

loss over the link.

Minor comments:

Comment #1:

Line 127: Figure 2a caption mentions inset (the amplified picture of the E-O modulation transfer curve when bias voltage ranging from -2 V to -0.5 V) which is absent in the figure itself. Please insert it.

Reply:

Comment #2:

Line 130: word signal is missing after word electrical.

Comment #3:

Lines 127-132: 1312 nm wavelength is mentioned several times. Since all figures show results with this wavelength, please mention it just once in the caption.

Comment #4:

Line 224: In Figure 4a there is a TL acronym, but in figure caption it is mentioned as TSL (tunable semiconductor laser). Please keep consistency.

Reviewer #2 (Remarks to the Author):

The authors have made much efforts to increase the modulation speed of silicon MRR via device design, modulation as well as signal processing. The key novelties they claim include two points, i.e. theoretical analysis of modulation nonlinearity in Si-MRMs, as well as design a physics-inspired bidirectional gate recurrent unit (Bi-GRU) neural network. Finally, they demonstrated optical interconnections at data rate up to 302 Gbps, showing a 1.1-dB received optical power gain than traditional nonlinear equalizers with spectral-efficiency of modulated signal to 5.20 bit/s/Hz. Although the authors managed to provide solid experimental results, the connection between two novelties is not well revealed. The authors have tried to describe the motivation of constructing a physics-inspired AI network, which to be best of own understanding, is to take the real modulation process into consideration. This is very interesting and important, but unfortunately not well explained, at least a better way should be find to explain the connection between physics-inspired AI (Fig. 4c) and theoretical model, to aid understanding from people who are not familiar with processes of neuron networks. From another point of view, the novelty of this work is not sharp. The theoretical analysis of modulation nonlinearity serves as a tool to help developing advanced AI network, and therefore how the physics-inspired AI network is constructed and why it provide great performance should be elaborated. Apart from this general point, I has a few other suggestions that the authors may consider to improve the work. Among these issues, the most critical part is that, the current

manuscript has too many serious grammar errors, which obviously can not be acceptable for the publication in Communications Engineering. In conclusion, I suggest major revision of this work, both technically and grammatically.

1.Line 112-114: the authors claim that “the radius (R) and racetrack length of the ring resonator is 8 μm and 5 μm , respectively, to keep the Si-MRM working near the critical coupling region at the wavelength of 1310 nm.” In my opinion, the critical coupling condition of an MRR has no direct connection with the radius and racetrack length.

2.How was Fig. 2c measured? This should be described.

3.Line 210-211: what is “wavelength detuning”? What is “wavelength difference”?

4.Just as an concrete example, a serious grammar error occurs at line 206-207: Since equation (3) tells .. is decided by the V_k and optical response at last time step ... which in turn relates to ... and further previous optical response. The sentence is not complete.

Responses to reviewers' comments

We thank the two reviewers for their efforts in reading our manuscript (COMMS-22-0309) titled “*300-Gbps Silicon Microring Modulator-based Optical Interconnection Accelerated by a Physics-inspired Neural Network*” and appreciate their inspiring and constructive comments. Based on the reviewers' suggestions, we almost rewrite our Manuscript and Supplementary Information to be more coherent and readable. We believe that the revision has been significantly improved and hopefully the additional data can further convince the reviewers. In order to clearly present the revision, the modifications according to the reviewers' comments are highlighted in red in revised Manuscript and Supplementary Information. Our point-to-point responses to the reviewers are listed as follows and marked in blue words together with the figures and tables labelled as Re.

Comments from Reviewer #1

In this paper, the authors showed equalization method with physics-inspired neural network to improve the spectral efficiency of Si micro-ring modulator. Proposed signal processing method mitigates modulation non-linearity-driven waveform distortions at the receiver side. By using this technique authors have achieved 302 Gbps transmission speed.

The authors presented some interesting approach and results. However, they might want to rewrite the paper or present the results in a different way.

Response: We thank the reviewer for the positive comments on our work. We have almost rewritten the whole paper, making the article more coherent and readable. Additionally, we have made detailed responses to the reviewer's concerns as seen in the following replies.

1. There is inconsistency between wavelength used in demonstration and spectrum measurement in supplementary information. 1312nm is used in whole demonstration, but in the Fig.S1 there is no spectrum result on 1312nm. In Fig.S1, authors showed resonance near 1310nm and based on their device description, next resonance should be around 1317nm. These resonance peaks are not closer to 1312nm at all, and this is big misleading. I understand this might coming from the fact that Fig.S1 is simulation. Personally, I would recommend adding measured spectrum directly in manuscript.

Response: We thank the reviewer for this comment. The original purpose of the simulation results in Fig.S1 is to explain why the laser wavelengths and bias voltages are two parameters that need iterative optimization from the perspective of spectrum. In fact, this can be explained directly through the modulation transfer curve measured by experiments (Fig. Re 1-1). Three modulation regions, named linear modulation region (I), nonlinear modulation region (II) and high-extinction-ratio modulation region (III), are highlighted in the Fig.5 (a). When increasing the bias voltage from I to III, despite the extinction ratio is enlarged, the modulation nonlinearity becomes more serious, and the RoP decreases. There exists a trade-off among extinction ratio, RoP and modulation nonlinearity. Therefore, the optimal bias voltage needs iterative optimization in experiments. The essence to affect the modulation characteristics of bias voltage is that it changes the wavelength detuning. The laser wavelength also change the wavelength detuning, and therefore it also needs iterative optimization.

To avoid the misleading caused by the wavelength's mismatch between simulations and experiments, we decide to delete the Fig.S1 and add additional analysis to the main manuscript directly. We did the following revision:

(1) the Fig.2 (a) is revised to Fig. Re 1-1.

Fig. Re 1-1: E-O modulation transfer curve (output optical power versus bias voltage). Inset: the amplified picture of the E-O modulation transfer curve when bias voltage ranging from -2 V to -0.5 V. I: linear modulation region. II: Nonlinear modulation region. III: High-extinction-ratio modulation region.

(2) The related analyses are rewritten.

“In fact, the working parameters of Si-MRMs: the bias voltage and input laser wavelength will jointly affect multiple properties of the modulated optical signal, including modulation amplitude, nonlinearity and output optical power according to the analysis in **Supplementary Information Fig. S1**. Hence, the optimization on the working parameters needs iterative search by experiments. The working parameters bringing the highest achievable data rate under a certain forward-error-correction threshold (FEC) is the optimal parameters...”

is revised to:

“We manually divide the modulation region where the bias voltage can be set into three regions: linear region (I), nonlinear region (II) and high-extinction-ratio (high-ER) region (the insets in Fig.2 (a)). There exists a trade-off among modulation characteristics (RoP, ER, modulation bandwidth and nonlinearity) at different modulation regions. Given that the carrier-depletion mode can achieve larger E-O modulation bandwidth than the carrier-injection mode [25], applying a negative bias voltage (Region I) for signal loading is better than a positive one for high-speed signal modulation. Furthermore, Region I also has the highest RoP, though its ER is lower than other regions. For short-reach optical interconnection scenarios where no optical amplifiers are applied, the optical signal with a higher RoP is crucial for higher data rate similar as ER. Hence, the bias voltage needs to be optimized by parameters iteration in experiments to balance the trade-off among these signal modulation characteristics. The combined effects of these characteristics eventually determine how fast the optical interconnection can be achieved. Since the essence to affect the modulation characteristics of bias voltage is that it changes the wavelength detuning, the laser wavelength, which changes the wavelength detuning as well, also needs iterative optimization.”

2. Line 158, authors mention optimal bias voltage of -0.9 V, with one level $V_{pp}=0.2$ V and 3-level $V_{pp}=0.6$ V (mentioned in Fig. 2 caption). However, in Fig. 2a we see that curve is quite flat around -0.9 V. It makes more sense to modulate around +0.9 V, since this value is on the slope of curve, which will produce much higher extinction ratio. Is x-axis (bias) any how reverted in graph Fig. 2a? Authors also stress they drive modulator in reverse bias (depletion mode). Sketch below explains the concern:

Response: We thank the reviewer for this comment. If considering the application scenarios where optical amplifiers are applied, it makes sense that higher extinction ratio results in higher speed or lower BER under the same average received optical power. However, in our application scenario (the short-reach optical interconnection) where no optical amplifiers will be applied, the received optical power can bring much more gain in transmission speed than extinction ratio. This fact is found in our experiment, the achievable data rate starts to decrease as increasing the bias voltage from -0.9 to 0, even higher (Supplementary Information Fig. S1). We deduce that two reasons lead to the decrease of data rate. First one is the steep drop of received optical power and the overlapping of AC signal with the static nonlinear modulation region of the transfer curve. The second one is the E-O bandwidth of Si-MRMs applied positive bias voltage is theoretically narrower than that applied positive bias voltage. Hence, the optimal bias voltage that can obtain the highest data rate is -0.9 V through parameters iteration in experiments.

The x-axis is not reverted in graph Fig.2a. we revise all “reverse bias voltage” in the main text to “negative bias voltage”.

We conduct the following revise,

“Meanwhile, sufficient optical power for receiver’s electro-optic response (>6.9 dBm) is guaranteed applying a reverse bias voltage. In fact, the working parameters of Si-MRMs: the bias voltage and input laser wavelength will jointly affect multiple properties of the modulated optical signal, including modulation amplitude, nonlinearity and output optical power according to the analysis in **Supplementary Information Fig. S1**. Hence, the optimization on the working parameters needs iterative search by experiments. The working parameters bringing the highest achievable data rate under a certain forward-error-correction threshold (FEC) is the optimal parameters. In our experiment, the optimal laser wavelength and bias voltage are 1312 nm and -0.9 V, respectively, according to the experimental results in **Supplementary Information Fig. S2**. At the optimal parameters, the measured S21 parameter is shown in Fig.2 (b). The -3-dB electrical bandwidth is around 42.5 GHz (measurement method is seen in **Methods**)”

is revised to

“We manually divide the modulation region where the bias voltage can be set into three regions: linear region (I), nonlinear region (II) and high-extinction-ratio (high-ER) region (the insets in Fig.2 (a)). There exists a trade-off among modulation characteristics (RoP, ER, modulation bandwidth and nonlinearity) at different modulation regions. Given that the carrier-depletion mode can achieve larger E-O

modulation bandwidth than the carrier-injection mode [25], applying a negative bias voltage (Region I) for signal loading is better than a positive one for high-speed signal modulation. Furthermore, Region I also has the highest RoP, though its ER is lower than other regions. For short-reach optical interconnection scenarios where no optical amplifiers are applied, the optical signal with a higher RoP is crucial for higher data rate similar as ER. Hence, the bias voltage needs to be optimized by parameters iteration in experiments to balance the trade-off among these signal modulation characteristics. The combined effects of these characteristics eventually determine how fast the optical interconnection can be achieved. Since the essence to affect the modulation characteristics of bias voltage is that it changes the wavelength detuning, the laser wavelength, which changes the wavelength detuning as well, also needs iterative optimization. The experimental results of the parameters iteration (**Supplementary Information Fig. S1**) show that the optimal bias voltage is -0.9 V at the optimal laser wavelength of 1312 nm. The input laser power is 15 dBm, the output optical power of the Si-MRM is >6.9 dBm at the optimal parameters. The measured S21 parameter is shown in Fig.2 (b), from which the -3-dB E-O bandwidth is around 42.5 GHz (measurement method is seen in **Methods**).”

3. I would like to recommend adding pam4 eye-diagram in the manuscript if they have in addition to Fig.5. Would be more impactful to show how eye-diagram can be improved through modified Bi-GRU.

Response: We thank the reviewer for this comment. We add the PAM4 eye-diagrams before and after the equalization of Bi-GRU as insets in Fig. 5 a-b. The revised Fig.5 is given as follows:

Fig. Re 1-2 Equalization performance of the modified Bi-GRU neural network for PAM-4 and DMT received signal. (a) The waveforms of transmitted (Tx) and received (Rx) PAM-4 signal at the sample rate of 5-GBaud. After the equalization of the modified Bi-GRU neural network on the received signal, the corresponding waveforms (red line) are shown in (b). Insets are their eye-diagrams. For 120-GBaud DMT signal, the waveforms of transmitted, received and equalized signal are shown in (c-d).

4. Please add PRBS length information. PRBS7 vs PRBS31 can make quite different result in BER measurement. Also I would like to recommend to add optical power and loss over the link.

Response: We thank the reviewer for this comment. We add the information of the PRBS that we used in experiments. The PRBS is generated by Mersenne Twister algorithm that has the period of $2^{19937} - 1$. This information is added in Line 245: “. . . Three patterns of PRBS are generated by Mersenne Twister algorithm [31] that has the period of $2^{19937} - 1$...”. In Method, we also add this information in Line 338: “In the first step, we generate a random PRBS (period: $2^{19937} - 1$) based on the Mersenne-twister algorithm”

Additionally, we added the optical power and loss over the link. The laser’s optical power is 15 dBm, and the RoP is around 7 dBm. Therefore, the loss is about 8 dB. This information is added in Line146: “...The input laser power is 15 dBm, the output optical power of the Si-MRM is >6.9 dBm at the optimal parameters...”

5. Line 127: Figure 2a caption mentions inset (the amplified picture of the E-O modulation transfer curve when bias voltage ranging from -2 V to -0.5 V) which is absent in the figure itself. Please insert it.

Response: We thank the reviewer for this comment. We have added the absent inset in Figure. 2a. The revised Figure.2a is shown here:

Fig. Re 1-3. Modulation characteristics of the Si-MRM. (a) E-O modulation transfer curve (output optical power versus bias voltage). Inset: the amplified picture of the E-O modulation transfer curve when bias voltage ranging from -2 V to -0.5 V. I: linear modulation region. II: Nonlinear modulation region. III: High-extinction-ratio modulation region. (b) The -3dB E-O bandwidth at the bias voltage of -0.9 V. (c) Static modulation nonlinearity indicated by the measured Amplitude/amplitude curve of transmitted (Tx) and received (Rx) electrical (bias voltage is -0.9 V and V_{pp} of signal is 0.6 V). (d) The normalized waveforms of 5-GBaud PAM-4 transmitted (Tx) and received (Rx) signals (bias voltage is -0.9 V and V_{pp} of signal is 0.2 V). The mazarine circle marks the overshoot phenomenon, i.e., the transient modulation nonlinearity. Laser wavelength in all figures is 1312.0 nm.

6. Line 130: word signal is missing after word electrical.

Response: We thank the reviewer for this comment. We have corrected this mistake.

7. Lines 127-132: 1312 nm wavelength is mentioned several times. Since all figures show results with this wavelength, please mention it just once in the caption.

Response: We thank the reviewer for this comment. We have revised the caption of Figure 2.

8. Line 224: In Figure 4a there is a TL acronym, but in figure caption it is mentioned as TSL (tunable semiconductor laser). Please keep consistency.

Response: We thank the reviewer for this comment. We have corrected this mistake.

Comments from Reviewer #2

The authors have made much efforts to increase the modulation speed of silicon MRR via device design, modulation as well as signal processing. The key novelties they claim include two points, i.e. theoretical analysis of modulation nonlinearity in Si-MRMs, as well as design a physics-inspired bidirectional gate recurrent unit (Bi-GRU) neural network. Finally, they demonstrated optical interconnections at data rate up to 302 Gbps, showing a 1.1-dB received optical power gain than traditional nonlinear equalizers with spectral-efficiency of modulated signal to 5.20 bit/s/Hz. Although the authors managed to provide solid experimental results, the connection between two novelties is not well revealed. The authors have tried to describe the motivation of constructing a physics-inspired AI network, which to be best of own understanding, is to take the real modulation process into consideration. This is very interesting and important, but unfortunately not well explained, at least a better way should be find to explain the connection between physics-inspired AI (Fig. 4c) and theoretical model, to aid understanding from people who are not familiar with processes of neuron networks. From another point of view, the novelty of this work is not sharp. The theoretical analysis of modulation nonlinearity serves as a tool to help developing advanced AI network, and therefore how the physics-inspired AI network is constructed and why it provide great performance should be elaborated. Apart from this general point, I has a few other suggestions that the authors may consider to improve the work. Among these issues, the most critical part is that, the current manuscript has too many serious grammar errors, which obviously can not be acceptable for the publication in Communications Engineering. In conclusion, I suggest major revision of this work, both technically and grammatically.

Response: We appreciate the reviewer for clearly illustrating our work and providing the constructive comments. The reason why our physics-inspired NN can equalize the nonlinear distortion brought by Si-MRMs is that the NN has a similar time-relevant model as the modulation model of Si-MRMs. To better explain this similarity between physics-inspired AI (Fig. 4c) and theoretical model, we almost rewrite the related narrative of Fig.4c to make it more readable. Here are the revision as follows:

“Rather than a “black box” application of AI models driven by hundreds of samples, the inner configuration of our modified Bi-GRU neural network is explainable and matches the modulation model of Si-MRMs. As shown in Fig.4 (c), the modulation model of a Si-MRM is that the k_{th} output optical signal O_k has a time-relevant pattern dependency on previous optical responses O_{k-1} and the k_{th} input electrical signal V_k , given as $O_k = \mathfrak{S}(O_{k-1}, V_k)$ in Equation (3), where $\mathfrak{S}(\cdot)$ is the dynamic nonlinear response of a Si-MRM. Referred to the equalization mechanism of traditional Volterra filters,

an equalizer having a similar model with the physic system can effectively approximate the inversion of the system by the fixed-point approach [30]. Here, our proposed forward GRU just provides such a similar time-relevant model as the modulation model of a Si-MRM. In the inner structure of the forward GRU, the $(k-i)$ th memory state C_{k-i}^f depends on previous memory states C_{k-i-1}^f and the $(k-i)$ th input sample O_{k-i} , expressed as $C_{k-i}^f = \mathfrak{N}(C_{k-i-1}^f, O_{k-i}) (0 \leq i \leq d)$, where $\mathfrak{N}(\cdot)$ is the inner nonlinear structure of a GRU cell (**Supplementary Information Note.2**). Therefore, the forward GRU theoretically has an effective nonlinear equalization ability to the modulation nonlinearity of Si-MRMs. Except for the nonlinear distortions, the mitigation of the inter-symbol interference effect resulting from the bandwidth limitation of Si-MRM and other E-O devices, requires the equalizer to consider the influence of symbols in future time slots. Hence, a backward GRU cell involved with the received symbols from k th to $(k+d)$ th time slots is merged with the forward GRU into the final modified-Bi-GRU. The effective nonlinear and linear equalization performance of the modified Bi-GRU for both PAM-4 and BPL-DMT signal is demonstrated experimentally shown in Fig. 5.”

The Fig.4 is also modified to Fig. Re 2-1:

Fig. Re 2-1 The proposed AI-accelerated Si-MRM-based optical transmission framework. (a). The proof-of-concept experimental system to realize high-speed short-reach optical interconnection. (AWG: arbitrary waveform generator; EA: electrical amplifier; DC: direct current; TL: tunable semiconductor laser; SSMF: standard single-mode fiber; Att.: attenuator; PD: photodetector; OSC: oscilloscope).

(b). The modified bidirectional gate recurrent unit (Bi-GRU) neural network is used to reconstruct the transmitted signal (Tx signal) from the received signal (Rx signal) captured from the OSC. (A: Amplitude). (c) The principle of Bi-GRU serving as a post-equalizer and the inner configuration of a GRU cell.

The point-to-point responses to the concerns are provided as seen in the following replies. We hope our additional experimental data and further elaborations will convince the reviewer.

1. Line 112-114: the authors claim that “the radius (R) and racetrack length of the ring resonator is 8 μm and 5 μm , respectively, to keep the Si-MRM working near the critical coupling region at the wavelength of 1310 nm.” In my opinion, the critical coupling condition of an MRR has no direct connection with the radius and racetrack length.

Response: We thank the reviewer for this comment. We apologize for the misleading written expression here. To design a Si-MRM with a high extinction ratio for high-quality O-band E-O modulation, we need keep the Si-MRM working at the critical coupling point. To make it, we need increase its ring-bus coupling efficiency at the wavelength of 1310 nm to equal the round-trip loss, by tuning two device parameters: the racetrack length and ring-bus gap.

Therefore, we revise the sentence “the radius (R) and racetrack length of the ring resonator is 8 μm and 5 μm , respectively, to keep the Si-MRM working near the critical coupling region at the wavelength of 1310 nm” to “The Si-MRM consists of a bus waveguide and a racetrack ring resonator with the radius (R) of 8 μm . To keep the Si-MRM working near the critical coupling region at the wavelength of 1310 nm, the racetrack length and the ring-bus gap width are respectively set to 5 μm and 280 nm to increase the coupling efficiency.”

2.How was Fig. 2c measured? This should be described.

Response: We thank the reviewer for this comment. We add the description of the measurement of AM/AM curve in **Methods**.

The added content in Methods is: “A typical characterization of the static E-O modulation nonlinearity of Si-MRMs is measuring AM/AM curve, in which the x-axis is the amplitude of the transmitted electrical signal sequence sending to AWG, the y-axis is the amplitude of the correspondingly received electrical signal that captured by OSC and then directly resampled to the same baud rate with the transmitted signal. The transmitted signal is a 5-Gbaud PAM-4 modulated digital signal. Both the transmitted and received electrical signals are normalized to [-1,1]. If there is no modulation nonlinearity during the E-O conversion of the Si-MRM, the AM/AM curve will present a linear trend. Otherwise, the curve will be nonlinear. It’s noted that the received optical power is adjusted to avoid the nonlinear region of PD.”

The added content in main text is : “Measuring the amplitude/amplitude (AM/AM) curve of transmitted and received signals is a typical method to show the static nonlinearity of a communication system (The measurement method is given in **Methods**).”

3.Line 210-211: what is “wavelength detuning”? What is “wavelength difference”?

Response: We thank the reviewer for this comment. “wavelength detuning” is the wavelength difference ($\Delta\lambda$) between the laser’s optical wavelength (λ_{laser}) and the resonance wavelength of a Si-MRM (λ_0), given as $\Delta\lambda = \lambda_{laser} - \lambda_0$.The resonance wavelength of a Si-MRM can be tuned by applying different external electric field and heat field of different magnitudes. The optical wavelength of the input light into a Si-MRM can be adjusted when using a tunable laser source. We unify all the “wavelength detuning” and “wavelength difference” in the main text to “wavelength detuning”.

In the first place where the “wavelength detuning” appears, we revise the description of wavelength detuning from “...wavelength detuning that defined as the wavelength difference between the laser wavelength and the resonance wavelength of the Si-MRM...” to “...wavelength detuning ($\Delta\lambda = \lambda_{laser} - \lambda_0$) that defined as the wavelength difference between the laser wavelength (λ_{laser}) and the resonance wavelength of the Si-MRM (λ_0)...”

4. Just as an concrete example, a serious grammar error occurs at line 206-207: Since equation (3) tells .. is decided by the V_k and optical response at last time step ... which in turn relates to ... and further previous optical response. The sentence is not complete.

Response: We thank the reviewer for this comment. We have conducted a thorough text check of the entire manuscript and have corrected various grammatical errors.

REVIEWERS' COMMENTS:

Reviewer #1 (Remarks to the Author):

In this revised paper, the authors went through all provided comments/feedback and fixed manuscript accordingly.

Main novelty of their work is understanding mechanism of nonlinear impairments and their time relevant dependency, and eventually developing the connection between the physical model and AI.

As proof-of-concept they successfully demonstrate this equalization method with physics-inspired neural network can improve the spectral efficiency of Si micro-ring modulator up to 302Gbps.

Also they made more effort to describe more details about their physics-inspired AI network itself.

I believe this manuscript is now suitable for the publication in Communications Engineering.